# Comparison of Transcriptomic Changes in Survivors of Exertional Heat Illness with Malignant Hyperthermia Susceptible Patients

**DOI:** 10.3390/ijms242216124

**Published:** 2023-11-09

**Authors:** Leon Chang, Lois Gardner, Carol House, Catherine Daly, Adrian Allsopp, Daniel Roiz de Sa, Marie-Anne Shaw, Philip M. Hopkins

**Affiliations:** 1Leeds Institute of Medical Research at St James’s, University of Leeds, Leeds LS9 7TF, UK; l.chang@leeds.ac.uk (L.C.); m.shaw@leeds.ac.uk (M.-A.S.); 2Survival and Thermal Medicine Department, Institute of Naval Medicine, Alverstoke, Hampshire PO12 2DL, UK; 3Malignant Hyperthermia Unit, St James’s University Hospital, Leeds LS9 7TF, UK; c.l.daly@leeds.ac.uk

**Keywords:** exertional heat illness, malignant hyperthermia, transcriptomics, inflammation

## Abstract

Exertional heat illness (EHI) is an occupational health hazard for athletes and military personnel–characterised by the inability to thermoregulate during exercise. The ability to thermoregulate can be studied using a standardised heat tolerance test (HTT) developed by The Institute of Naval Medicine. In this study, we investigated whole blood gene expression (at baseline, 2 h post-HTT and 24 h post-HTT) in male subjects with either a history of EHI or known susceptibility to malignant hyperthermia (MHS): a pharmacogenetic condition with similar clinical phenotype. Compared to healthy controls at baseline, 291 genes were differentially expressed in the EHI cohort, with functional enrichment in inflammatory response genes (up to a four-fold increase). In contrast, the MHS cohort featured 1019 differentially expressed genes with significant down-regulation of genes associated with oxidative phosphorylation (OXPHOS). A number of differentially expressed genes in the inflammation and OXPHOS pathways overlapped between the EHI and MHS subjects, indicating a common underlying pathophysiology. Transcriptome profiles between subjects who passed and failed the HTT (based on whether they achieved a plateau in core temperature or not, respectively) were not discernable at baseline, and HTT was shown to elevate inflammatory response gene expression across all clinical phenotypes.

## 1. Introduction

Exertional heat illness (EHI) causes injury and death through the effects of salt and water depletion, direct thermal tissue damage, or a combination of these. It results from a failure to dissipate the heat generated by intense and sustained metabolic activity of voluntary muscles [1]. It occurs especially in endurance sporting events [2] and during military training and exercises [3]. Although a number of “environmental” risk factors have been identified (e.g., climatic conditions, food or water deprivation, obesity, lack of physical fitness, concurrent viral or dehydrating illness, recent alcohol or recreational drug use, some prescribed medicines) [3], there remain individuals with a clear disposition to develop heat illness (repeated occurrences in cool climates) [4,5].

Vanuxem et al. measured exercise capacity and muscle energy metabolism in military personnel recovering from EHI episodes, featuring hyperthermia (≥40 °C), neurological impairment and rhabdomyolysis [6]. An EHI group had lower VO_2_ max scores and lower maximal workloads than healthy controls during maximal exercise on a cycloergometer, more than 5 months after they presented with EHI. They also demonstrated a limited thermoregulatory capacity, with higher body temperature post-exercise than the healthy controls. Plasma-free fatty acids, glycerol and blood lactate concentrations were significantly higher in the EHI patients during and after exercise, suggesting increased lipolysis and impairment of the oxidative phosphorylation (OXPHOS) pathway.

Based on in vitro skeletal muscle pharmacological contracture tests (IVCT) that are used to diagnose susceptibility to malignant hyperthermia (MH, a progressive hypermetabolic, hyperthermic reaction with similar clinical features to EHI but which is triggered in genetically susceptible individuals by commonly used anesthetic drugs [7]) we identified a skeletal muscle abnormality in two survivors of EHI and their immediate blood relatives [5]. Of subsequent EHI patients who we have characterised by IVCT, 35% demonstrated abnormal contracture responses [8]. Our EHI cohort consists mainly of individuals who have sustained one or more episodes of EHI during military duties. These military cases were referred for IVCT because of their clinical history and because they repeatedly were unable to demonstrate thermoregulation during a standardised heat tolerance test (HTT) [9].

Genetic analyses of our EHI cohort have confirmed the role of *RYR1* (the gene principally implicated in susceptibility to MH) in the heritability of EHI but suggested genetic heterogeneity and a multigenic model. We found non-polymorphic potentially pathogenic variants in genes encoding proteins involved in skeletal muscle calcium homeostasis, oxidative metabolism and membrane excitability [8]. While it is possible that some of these variants have potential for use in pre-symptomatic screening for risk of EHI, it is likely that further genes will be discovered to play a role.

One approach to elucidating molecular and cellular mechanisms of EHI and underlying genetic defects is to identify differential gene expression affecting individual genes and biological pathways or processes between survivors of EHI and controls. Here we report whole blood gene expression before (basal) and after the subjects undertook a standardised HTT. We also included a group of MH-susceptible volunteers to examine any similarities in gene expression that might define common underlying molecular defects between the two disorders, and their responses to exertional heat stress. We aimed to explore the possibility of using differential gene expression as a biomarker for the risk of EHI, MH susceptibility, or both.

## 2. Results

### 2.1. Subject Characteristics and HTT Data

The details of the subjects from the first phase of recruitment and their HTT data have been published previously [9]. Those of the subjects recruited in the second phase are presented in the Appendix A.

### 2.2. Analysis of First Sample Set Phenotypes at Baseline

This analysis compared differences in gene expression in blood samples taken prior to heat tolerance testing and included five healthy controls, all passing the HTT, six EHI cases, one of whom failed the HTT and six MH susceptible individuals, one of whom failed the HTT.

### 2.3. Exertional Heat Illness at Baseline

Relative to control samples, 291 genes (153 upregulated/138 downregulated) were differentially expressed in the EHI cohort. Enrichment analysis of upregulated genes highlighted increased activity in the “Interferon Gamma Response”, “TNF-alpha Signalling via NF-kB” and “Inflammatory Response” hallmarks (Figure 1A). Several genes appeared in multiple hallmarks and were found upregulated up to four-fold above healthy controls at baseline (Figure 1B). In contrast, downregulated genes were only enriched in the “Myc Targets V1” hallmark (Figure 1C). No ontology terms were significantly enriched in upregulated genes, but the downregulated gene-set was annotated into a range of biological process (GO:BP) terms associated with telomerase RNA and protein localisation (Table 1).

### 2.4. Malignant Hyperthermia at Baseline

A greater divergence in transcriptome profile was observed in the MHS cohort featuring 1019 genes differentially expressed compared to controls (575 upregulated/444 downregulated). Upregulated genes were significantly enriched in ‘TGF-beta Signalling’, ‘Mitotic Spindle’ and ‘UV Response Dn’ hallmarks (Figure 1D), with ontology terms associated mainly with transcriptional regulation (Appendix A). The downregulated gene-set was most highly enriched in the ‘Oxidative Phosphorylation’ hallmark with 30 genes found in this category. This is supported by the ontology findings which highlight many GO:BP, molecular function (GO:MF) and cellular component (GO:CC) results surrounding mitochondrial functions—with emphasis on complex I NADH dehydrogenase activity (Appendix A).

### 2.5. Exertional Heat Illness Compared to Malignant Hyperthermia

To identify similarities between both conditions a Venn diagram was created comparing both ‘EHI vs. Control’ and ‘MHS vs. Control’ baseline gene sets (Figure 2A). EHI and MHS samples share a total of 84 genes, with 51 upregulated and 33 downregulated in both conditions. Upregulated genes were associated with the ‘Interferon Gamma Response’, ‘TNF-alpha Signalling via NF-kB’ and ‘Inflammatory Response’ hallmarks whilst downregulated genes were enriched in ‘Oxidative Phosphorylation’ and ‘Myc Targets V1’. The genes allocated to these hallmarks were visualised on a heatmap which showed distinct clusters separating control samples to both MHS and EHI individuals (Figure 2B). A list of all genes found in each Venn diagram segment can be found in the Appendix A.

### 2.6. Response to Heat Tolerance Testing

The effect of the HTT on gene expression was explored at three different timepoints (baseline, 2 h, and 24 h) using EHI (*n* = 6), MHS (*n* = 6) and control (*n* = 5) samples. No significant interaction effect between phenotype (EHI/MH/control) and HTT timepoints was observed. Phenotype groups were subsequently combined for simple timepoint analysis revealing 5074 genes differentially expressed at 2 h vs. baseline and 7079 genes differentially expressed at 24 h vs. 2 h. Genes upregulated (2682) at the 2 h timepoint were mainly associated with the ‘TNF-α Signalling via NF-κB’ and ‘Inflammatory Response’ hallmarks, whilst downregulated genes (2392) were associated mainly with Myc and mTORC1 signalling. At 24 h, the effects on gene expression appear to normalize as the ‘TNF-α Signalling via NF-κB’ and ‘Inflammatory Response’ hallmarks are downregulated with Myc and mTORC1 signalling hallmarks upregulated, respectively (Appendix A). The genes and functional annotations described in this section will be referred to as the ‘normal’ 2 h and 24 h HTT responses.

### 2.7. Analysis of Combined Sample Sets

#### Heat Tolerance Test Response PASS vs. FAIL

Due to limitations in the initial sample set, a second analysis was performed after recruiting additional EHI samples from individuals who failed the HTT (baseline and 2 h). This allowed us to further compare EHI individuals who passed the HTT (EHI-PASS, *n* = 10) with those who failed (EHI-FAIL, *n* = 10). The variability between batches was accounted for by incorporating a batch term within the DESeq2 design model. Between-group comparisons at each time point showed few differences (Figure 3A). Clustering of baseline samples on the PCA plot shows no observable difference in baseline gene expression between EHI-PASS and EHI-FAIL. Likewise, 2 h HTT clusters are also overlapping, but show greater variability than baseline counterparts. Only one gene was differentially expressed at baseline (*BTNL3* downregulated in EHI-FAIL), and only four genes were found across the 2 h timepoint (*PDK4* upregulated; *BTNL3*, *PRDX2* and *TECTA* downregulated in EHI-FAIL).

Within-group analysis (before and after HTT) revealed 1586 (1227 upregulated/359 downregulated) and 3203 genes (2030 upregulated/1173 downregulated) differentially expressed in EHI-PASS and EHI-FAIL responses respectively. All the genes found in the EHI-PASS and EHI-FAIL responses were cross-compared using a Venn diagram which showed a large crossover, accounting for 87.1% of the EHI PASS response (1381 genes, 1137 upregulated and 244 downregulated) (Figure 3B). However, this overlapping region only occupied 43.1% of the total EHI-FAIL response with an additional 1822 genes differentially expressed uniquely to EHI-FAIL samples (893 upregulated and 929 downregulated).

Subsequent enrichment analysis of upregulated genes showed associations with ‘TNF-alpha Signalling via NF-kB’, ‘PI3K/AKT/mTOR Signalling’, ‘IL-6/JAK/STAT3 Signalling’ and ‘Apoptosis’, all of which were also enriched in the normal 2 h HTT response previously described (Figure 3C). Enrichment of downregulated genes featured ‘Myc Targets V2’, ‘Myc Targets V1’, ‘E2F Targets’, ‘Oxidative Phosphorylation’ and ‘mTORC1 Signalling’ in the top five hallmarks (Figure 3D). These results are also in the top five terms of the normal 2 h HTT response, with the exception of ‘Oxidative Phosphorylation’ which has a higher ranking in EHI FAIL samples. An additional 28 genes involved with Oxidative Phosphorylation were downregulated uniquely to a failed HTT response.

## 3. Discussion

### 3.1. Baseline Comparisons

MHS and EHI baseline comparisons reveal dysregulated gene expression in metabolic pathways and immune response pathways. Comparisons of EHI and MHS samples with controls at baseline, along with responses shared between EHI and MHS samples, indicate increased expression of genes involved in responses to IFNγ and TNFα via NFΚB signalling, as hallmarks of a pro-inflammatory response. Although interestingly, the MHS versus control analysis also highlighted increased expression of genes associated with TGFβ signalling, typically regulating the pro-inflammatory response. In theory, specific genes of interest upregulated in EHI individuals might be detected by a simple blood test. However, the level of up-regulation and gene profile overlapping with other conditions would compromise sensitivity and specificity.

Differentially expressed genes associated with pro-inflammatory response pathways were expressed up to four-fold higher in subjects with EHI, compared to healthy controls at rest. Dysregulation of the inflammation pathway is not surprising as it is widely accepted that EHI, which advances to EHS triggers a systematic inflammatory response, which can lead to multi-organ failure and death in extreme cases [10,11]. However, this striking elevation of pro-inflammatory gene expression at baseline suggests that those with EHI are under greater physiological stress and may be more susceptible to excessive inflammatory responses following exercise and heat stimuli.

Together with the upregulated pro-inflammatory hallmarks in EHI subjects at baseline, the gene ontology analysis revealed biological processes associated with the maintenance of telomeres to be downregulated. Telomerase-related genes, involved in the maintenance of telomeres with roles in cell cycle control and aging/cell death are also involved in multiple pathways including immune response pathways relevant to the pro-inflammatory response [12,13]. Molecular studies on human EHI patients are limited, with the few existing studies finding it difficult to dissociate the EHI immune response from the normal effects of exercise and heat stress [14,15].

In addition to gene expression in immune response pathways, we observed decreased expression of MYC target genes involved in cell proliferation, including mitochondrial biogenesis, but also apoptosis [16]. Through its regulation of target genes, c-myc itself inhibits skeletal muscle myoblast differentiation and promotes proliferation [17]. With a mutation in Ryr1, pigs provided the original animal model for MH, and an epistatic interaction between c-myc and Ryr1 has been reported to be associated with carcass quality traits [18].

Genes associated with OXPHOS were downregulated most prominently in MH susceptible patients, although there also appears to be some effect in those with a history of EHI. Ontology results surrounding mitochondrial function had an emphasis on Complex I NADH dehydrogenase, the first component of the electron transport chain. Previously we have used high-resolution respirometry to compare oxygen consumption rates (oxygen flux) between permeabilised human MH susceptible and control skeletal muscle fibres. Baseline comparisons showed significantly increased mitochondrial mass (Complex IV, *p* = 0.021) but lower flux control ratios in ‘Complex I + Complex II _(OXPHOS)_’ and ‘Complex II _(ETS)_’ of mitochondria from MH susceptible individuals compared with control (*p* = 0.033 and 0.005, respectively) showing that human MH susceptible mitochondria have a functional deficiency [19]. In a study of the related condition, recurrent exertional rhabdomyolysis in race horses, a genome-wide RNAseq analysis of gluteal muscle suggested that susceptibility was associated with genes affecting myoplasmic Ca^2+^ regulation and mitochondria [20].

Adipogenesis and fatty acid metabolism genes are yet another interesting group of genes downregulated particularly in the MH susceptible versus control baseline comparison. In the mouse model of MH carrying a Y524S mutation in the Ryr1 gene, the mice have exacerbated brown fat adaptive thermogenesis and brown fat capacity [21]. Any genetic relationship between the control of brown adipose tissue and thermoregulation will necessarily be complex. A recent metabolomic study also found a significant accumulation of numerous fatty acid metabolites in human MHS muscle, indicating a potential defect in lipid metabolism [22].

The relationship between MH and EHI susceptibility is established with many confirmed EHI patients testing positive with the gold standard test for MH susceptibility, the IVCT [23,24]. However, the nature of this relationship is unclear. Gene expression profiling suggests a combination of heightened pro-inflammatory responsiveness, together with baseline mitochondrial dysfunction will play a part.

### 3.2. Response to the HTT and Timepoint Analyses

There are many differentially expressed genes between timepoints, but since these are not influenced by the EHI/MHS/control phenotype as evidenced by the absence of a significant interaction term, timepoint analysis was conducted using all phenotypes to provide ‘normal’ 2 h and 24 h responses to the HTT. The normal responses to the HTT tended to mirror the most significant changes seen in MH and EHI individuals compared to controls at baseline. At 2 h pro-inflammatory responses, i.e., genes associated with TNFα via NFΚB signalling were elevated, whereas genes associated with MYC and mTORC1 signalling were downregulated. At 24 h the opposite was observed, indicating a reversal of the acute gene expression changes triggered by the HTT.

There have been several transcriptomic studies looking specifically at responses in peripheral blood, up to 24 h after heat exposure. Participants in these studies were healthy and comparable to our control group, i.e., without reported episodes of EHI. Exposure was passive as occurs in classic heat stroke, or through exercise as occurs in EHS [25]. Endurance exercise is typically associated with an overall decrease in gene expression, but an increase in genes coding for the proinflammatory cytokines TNF-α, IL-1, IL-6 and IL-8 and paradoxically the anti-inflammatory IL-1ra (receptor antagonist), and some mitochondrial dysfunction [26,27,28]. Directionality of change may be affected differently in classic heat stroke compared with EHS [29].

### 3.3. Differential Gene Expression between EHI-PASS and FAIL Individuals at Each Timepoint

There were very few significant differences between EHI-PASS and EHI-FAIL groups at each timepoint and no interaction between outcome and the 2 h timepoint. At baseline, a single gene, *BTNL3*, was significantly downregulated in the EHI-FAIL group. At the 2 h timepoint *PDK4* was significantly upregulated and *BTNL3*, *PRDX2* and *TECTA* downregulated in the EHI-FAIL group. *BTNL3* encodes butyrophilin-like 3, a major immunoglobulin-like protein known to bind to T-cell receptors and regulate γδT cell subsets in both innate and adaptive immunity [30]. Pyruvate dehydrogenase kinase isoenzyme 4, *PDK4*, reversibly inactivates the mitochondrial pyruvate dehydrogenase complex, functioning as a primary regulator of respiration. Peroxide reductase 2, *PRDX2*, plays a role in the protection of red blood cells from oxidative stress generated by reactive oxygen species during normal metabolism. The *TECTA* gene product is specialised and not clearly relatable in the response to the HTT.

Within-group analyses showed that individuals who fail the HTT appear to exhibit greater changes in gene expression profile than those who pass, 2 h after testing. The HTT causes changes and up-regulation in stress/immune response pathways as might be expected, but it is interesting that there was an additional emphasis on IL-6/JAK/STAT3 signalling which was unique to the EHI-FAIL subjects. Pro-inflammatory cytokines such as IL-6, have been labelled as exercise-induced myokine, but a study has shown that IL-6 levels are significantly higher in human myotubes harbouring *RYR1* mutations, suggesting a direct link to Ca2+ homeostasis [31,32]. Our findings are similar to observations made using mouse models, where a skeletal muscle cytokine response, including IL-6, was detected in a mouse model of environmental heat stroke [33,34]. In a further study of EHS in mice Iwaniec et al. looked at the type and timing of release of acute phase proteins typically part of an inflammatory response and comparing skeletal muscle and liver revealed tissue specificity [35].

The downregulated genes in the EHI-FAIL grouping are enriched in the OXPHOS, adipogenesis, mTORC1 signalling and Myc categories. Hence, not only is there some dysregulation in these processes at baseline but they are also influenced by the HTT response disproportionately in the EHI-FAIL individuals. The sensitivity and specificity of the HTT have not been established due to the absence of a validated diagnostic test for EHI susceptibility, although there is evidence that it lacks complete sensitivity. There have been examples of EHI military personnel who effectively thermoregulate during the assessment but subsequently experience a further episode of EHI and are later shown to have an abnormal IVCT response [8].

A U.S. study by Ren et al. (2019) used a different study design where six individuals with a history of EHS and 18 controls were subjected to both an HTT and a thermoneutral test (on separate days) with blood collected before and immediately post-testing for genome-wide microarray analysis [36]. For each individual pre-test expression levels were subtracted from post-test levels, and changes from the thermoneutral test were subtracted from changes in the HTT. Genes relating to interleukins and cellular stress were downregulated in those participants with a history of EHS [36]. Interestingly, all of the patients with a history of EHS (most with multiple episodes) demonstrated normal thermoregulation during the HTT, while five of the 18 controls did not. The HTT used by Ren and colleagues, with high environmental heat stress but modest metabolic thermal load may, therefore, have failed to initiate dysregulated skeletal muscle responses to sustained physical work that may underlie the genetic predisposition to EHI. In contrast, permanent heat intolerance as defined by the INM HTT, is associated with abnormal IVCT responses in more than 30% of individuals and also with variants in genes encoding proteins involved in skeletal muscle Ca^2+^ regulation, membrane excitability and metabolism [8]. This and other differences in study protocols, potential confounders, timing of sampling and analysis make differences between the findings of Ren et al. and our current study difficult to interpret.

In summary, we have found differences in basal whole blood gene expression in patients susceptible to MH or with a history of EHI compared to controls, including an increase in pro-inflammatory gene expression and a decrease in expression of genes associated with OXPHOS, adipogenesis and myoblast proliferation. The HTT triggered gene expression changes in the same pathways observed in the baseline data, but these changes were more pronounced in those who failed the HTT, featuring a greater number of differentially expressed genes. We speculate that the elevated pro-inflammatory gene expression at baseline pre-disposes individuals to excessive inflammatory events under duress, resulting in episodes of EHI. There is potential for this to be used as a screening tool for EHI but more research is required to increase the specificity for this condition.

## 4. Methods

### 4.1. Subjects and Samples

All subjects were male, aged 18–40 years old, with a good base level of fitness. Recruitment and sample analysis took place in two phases. The first set of samples was composed of six members of the armed forces who had presented with EHI (EHI group), six healthy military personnel with no history of EHI (control group) and six MHS patients from the Leeds MH cohort MHS group. The MHS group was age, sex and fitness-matched to the military volunteers. The EHI recruits demonstrate a range of phenotypes, including collapse, loss of consciousness and hyperthermia. Individuals recruited into the MHS group were either MH index cases or family members, who all carried a pathogenic variant in the *RYR1* gene, and had a positive IVCT with 0.5 g contracture at 2% *v*/*v* halothane and a 0.2 g contracture at 2 mmol L^−1^ caffeine. An additional EHI volunteer was recruited due to technical issues obtaining core temperature and sweat measurements from a member of the original EHI group.

The second sample set was collected because only one member of the original EHI group demonstrated an inability to thermoregulate in the HTT and there was interest in comparing the gene expression profiles of EHI subjects who had failed the HTT (Inability to thermoregulate) with EHI subjects who had passed the HTT (demonstrated thermoregulation). We, therefore, recruited further military personnel who had presented with EHI, five who passed the HTT and nine who failed the HTT. Subject characteristics for this cohort are supplied in Appendix A.

All volunteers underwent the same standardised HTT [9], carried out by exercise physiologists in survival and thermal medicine at the Institute of Naval Medicine. In brief, the test was carried out in a heat-controlled chamber set to 34 °C (dry bulb temperature) with 40% relative humidity, producing a wet bulb globe temperature (WBGT) index of 27 °C. Each subject initially completed a VO2 max test. They were then exercised at 60% of their VO2 max on a treadmill in full military kit while carrying a 14 kg rucksack for the first 30 min. Their rucksack and jackets were removed as they continued to walk for a further 15 min. After 45 min, their t-shirts were removed and they continued until a plateau in rectal temperature was observed (HTT passed) or 90 min had passed (HTT failed). The HTT also failed if their rectal temperature reached 39.5 °C, in which case they were removed from the chamber and actively cooled. Measurements of heart rate, core (rectal and intestinal) and skin temperature, heat flow data, blood flow data, and sweat capsule data were recorded at 1 min intervals throughout the test.

Blood samples for global gene expression analysis were collected into PAXgene^®^ Blood RNA tubes (PreAnalytiX^®^ GmbH, Hombrechtikon, Switzerland) at three time points (baseline, 2 h and 24 h) for the first phase but only baseline and 2 h for the second phase. The samples were stored at −20 °C prior to transfer to the Leeds MH Investigation Unit to be processed.

### 4.2. Preparation of RNA

RNA was extracted from the whole blood samples using PAXgene^®^ Blood RNA kits (PreAnalytiX^®^ GmbH, Hombrechtikon, Switzerland) according to the manufacturer’s instructions. Each RNA sample was washed, purified, and treated with DNase 1 prior to storage at −80 °C. Purified RNA samples were quantified and quality checked using Agilent’s RNA ScreenTape assay and 2200 TapeStation instrument (Agilent technologies, Santa Clara, CA, USA) according to the manufacturer’s instructions. RNA samples with integrity scores of RIN^e^ > 7.0 were classed as good quality and acceptable for further use.

### 4.3. Truseq^®^ Stranded mRNA Library Preparation

Poly (A)-selected mRNA libraries were created from the isolated RNA by the Next Generation Sequencing (NGS) Facility at St James’s Hospital, Leeds. RNA samples were first quantified using the Qubit^TM^ RNA BR Assay Kit (Invitrogen, Carlsbad, CA, USA) before being used to create libraries using the Truseq Stranded mRNA library preparation kit (Illumina, San Diego, CA, USA), according to the manufacturer’s guidelines. All cDNA libraries were quality-checked using the Agilent D1000 ScreenTape (Agilent technologies, Santa Clara, CA, USA) and quantified using the Quant-iT^TM^ PicoGreen^®^ dsDNA assay (Invitrogen, Carlsbad, CA, USA).

### 4.4. Illumina^®^ HiSeq^®^ NGS

Equimolar concentrations of each mRNA-enriched cDNA library were pooled and sequenced across lanes of Illumina’s HiSeq^®^ 3000 platform (Illumina, San Diego, CA, USA), according to the manufacturer’s instructions. The 150 bp paired-end reads were de-multiplexed to produce a forward and reverse Fastq file for each sample, on each lane. The RNA sequencing for both runs was performed by the Leeds University NGS facility, located at St James’s University Hospital, Leeds. The average read depths from the first sample set (*n* = 51) and a second set (*n* = 28) of samples were 36.8 M and 25 M reads per sample, respectively.

### 4.5. Sample Size Calculation for RNA-seq

The ‘ssizeRNA’ package in R was used to calculate a basic estimation of the sample size required to reach a power of 0.8 [37]. This estimation assumes that all genes share the same average read count, equal dispersion and fold change across each group. The number of genes to be tested was set to 20,000, the proportion of non-differentially expressed genes as 0.8, the average read count for each gene as 50, dispersion as 0.1, fold-change as 2, the FDR level as 0.05 and the desired power as 0.8. This produced an estimated sample size of 9 for each group and the time point required to reach a power of 0.86.

### 4.6. Differential Gene Expression Analysis

Fastq files for each technical replicate were combined and trimmed using Cutadapt [38] before the quality assessment was performed using FastQC [39]. Files were aligned to the human reference genome GRCh38.p13 using the STAR aligner [40] before quantification using featureCounts [41]. The RStudio package DESeq2 was used to generate lists of differentially expressed genes [42].

### 4.7. Enrichment Analysis and Gene Ontology

Functional annotation of differentially expressed genes was achieved using Enrichr [43,44]. Initial enrichment analysis was performed using the 50 hallmark gene-sets from the Human Molecular Signatures Database (MSigDB, v2023.1.hs, San Diego, CA, USA). Hallmark gene sets summarize and represent specific well-defined biological states or processes. Gene ontology analysis was performed using gene-sets from the GO biological processes 2018, GO Molecular Function 2018 and GO Cellular Component 2018 libraries. Data visualisations were created using web-based scatterplot, clustergrammer and compressed bar chart appyters (https://appyters.maayanlab.cloud; accessed on 15 September 23) [45].

### 4.8. Statistics

Differentially expressed genes generated by DESeq2 are defined using a threshold of adjusted *p*-value < 0.05, adjusted for multiple testing using the Benjamini–Hochberg (BH) adjustment [46]. Significantly enriched pathway and ontology terms from Enrichr are defined with a *q*-value < 0.05.

## Figures and Tables

**Figure 1 ijms-24-16124-f001:**
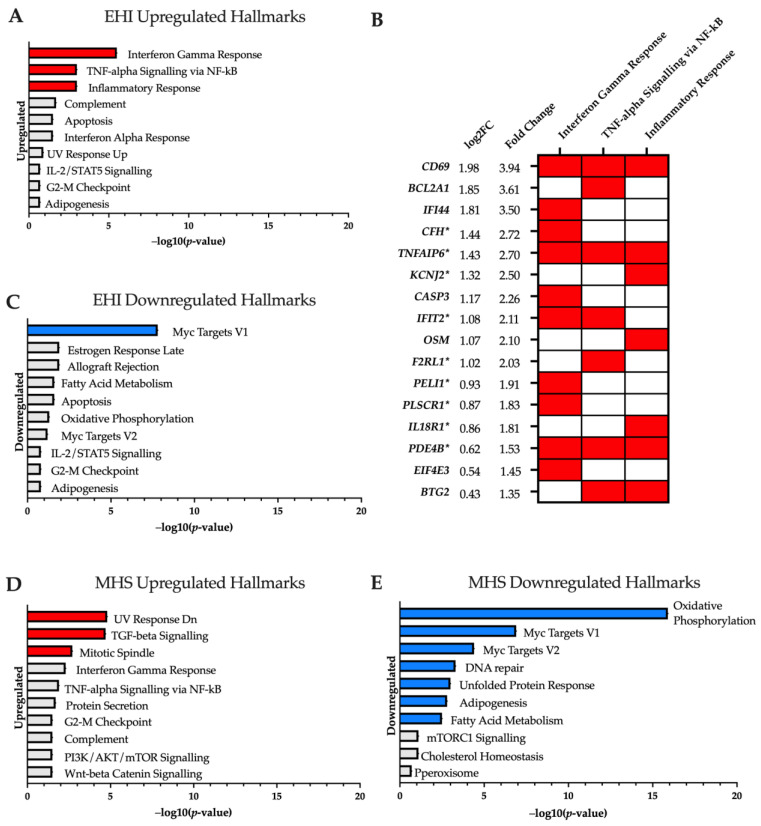
Significantly enriched MSigDB hallmarks (adjusted *p*-value < 0.05) for (**A**) ‘Control vs. EHI’ upregulated; (**C**) ‘Control vs. EHI’ downregulated; (**D**) ‘Control vs. MHS’ upregulated; (**E**) ‘Control vs. MHS’ downregulated; are highlighted in red (upregulation) and blue (downregulation) in −log10 (*p*-value) order. (**B**) Genes of interest upregulated in EHI samples at baseline, displayed in order of fold change. Red blocks indicate the associated enrichment hallmark category and asterisks * on Gene IDs indicate genes also upregulated in MHS vs. controls at baseline.

**Figure 2 ijms-24-16124-f002:**
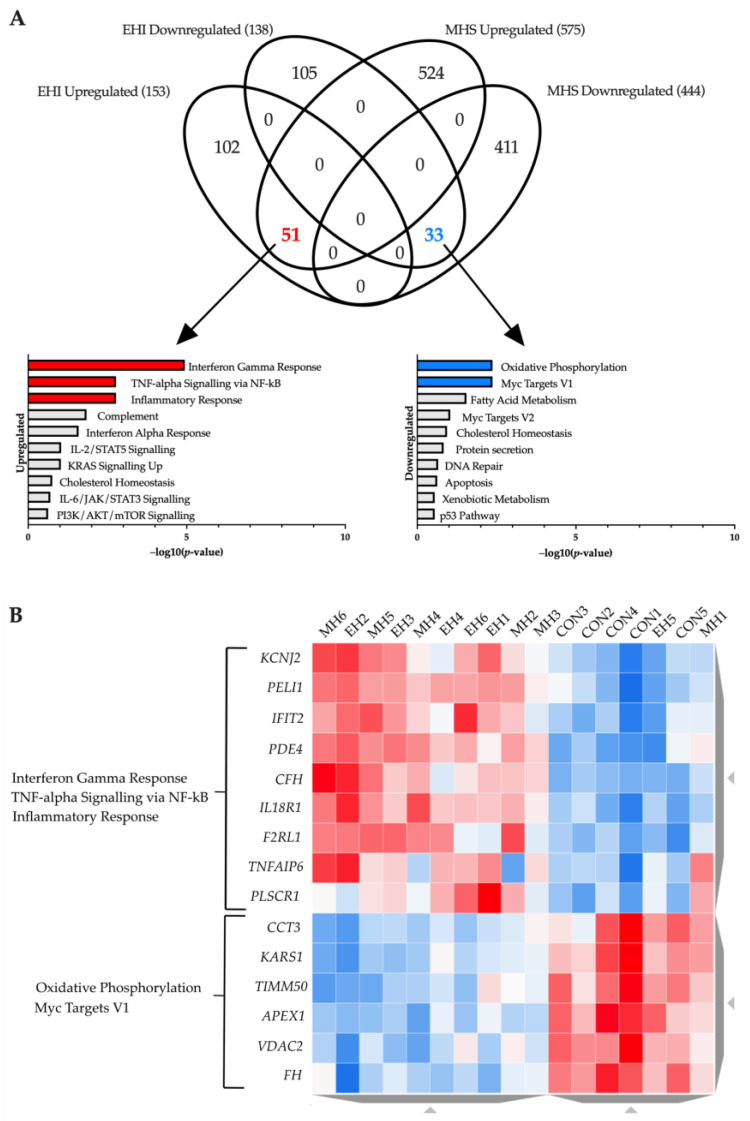
(**A**) Venn diagram illustrating a comparison of ‘Control vs. MHS’ and ‘Control vs. EHI’ differentially expressed gene-sets. Two overlaps can be seen with 51 genes upregulated, and 33 genes downregulated in both MHS and EHI samples when compared to control. Bar charts highlighting significantly enriched functional hallmarks (adjusted *p*-value < 0.05) are provided for each overlap in red (upregulation) and blue (downregulation) in −log10 (*p*-value) order. (**B**) Heatmap of baseline expression. The differentially expressed genes shared in both MHS and EHI samples within enriched hallmarks are displayed in clusters of relative expression levels. Red indicates high expression and blue reflects reduced expression; the intensity of the colours reflect the range in expression level.

**Figure 3 ijms-24-16124-f003:**
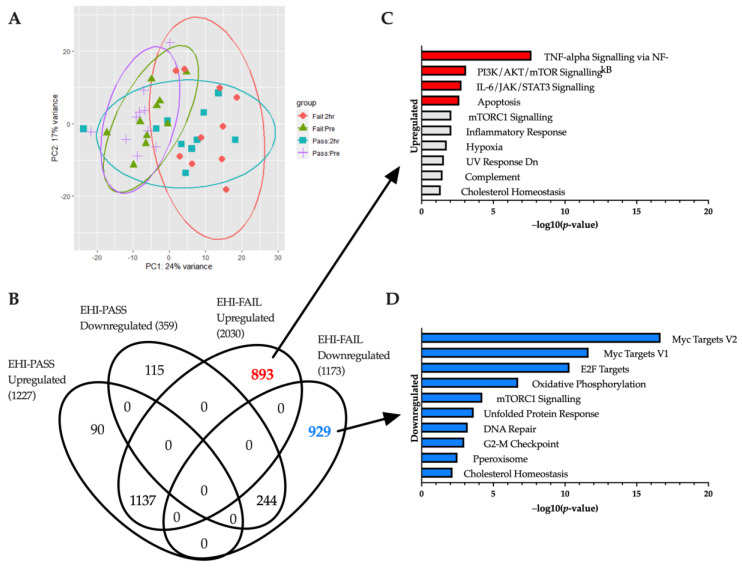
(**A**) Principal component analysis. Samples in the EHI cohort are displayed along two principal components to help illustrate the variance between groups. EHI samples were divided into those who passed HTT (*n* = 10) and those who failed (*n* = 10) and plotted with their Pre and 2 h post-HTT samples. (**B**) Genes unique to EHI-FAIL samples. EHI samples are divided into sub-groups of those who passed (EHI-PASS) and failed (EHI-FAIL) the heat tolerance test. This Venn diagram shows a comparison between the HTT treatment responses of EHI-PASS and EHI-FAIL individuals, revealing differentially expressed genes in common, and unique to each sub-group. 893 genes were upregulated uniquely to EHI-FAIL samples, whilst 929 were downregulated. Bar charts highlighting significantly enriched (**C**) upregulated and (**D**) downregulated hallmarks (adjusted *p*-value < 0.05) are provided for both segments.

**Table 1 ijms-24-16124-t001:** Gene ontology (Downregulated in EHI cohort). Downregulated gene ontology results for the EHI vs. Control baseline comparison. Statistically significant (*q*-value < 0.05) GO Biological Process, GO Molecular Function and GO Cellular Component terms are ordered by *q*-value, which corresponds to the *p*-value adjusted for multiple comparisons.

GO Biological Process Term	*p*-Value	*q*-Value
regulation of telomerase RNA localization to Cajal body (GO:1904872)	1.68 × 10^−9^	2.00 × 10^−6^
positive regulation of telomerase RNA localization to Cajal body (GO:1904874)	4.13 × 10^−8^	2.00 × 10^−5^
positive regulation of establishment of protein localization to telomere (GO:1904851)	4.41 × 10^−7^	1.10 × 10^−4^
regulation of establishment of protein localization to telomere (GO:0070203)	6.90 × 10^−7^	1.10 × 10^−4^
regulation of protein localization to Cajal body (GO:1904869)	6.90 × 10^−7^	1.10 × 10^−4^
positive regulation of protein localization to Cajal body (GO:1904871)	6.90 × 10^−7^	1.10 × 10^−4^
positive regulation of protein localization to chromosome, telomeric region (GO:1904816)	1.03 × 10^−6^	1.41 × 10^−4^
tRNA aminoacylation (GO:0043039)	6.08 × 10^−6^	7.29 × 10^−4^
positive regulation of establishment of protein localization (GO:1904951)	7.76 × 10^−6^	8.28 × 10^−4^
ncRNA processing (GO:0034470)	1.10 × 10^−5^	9.89 × 10^−4^
**GO Molecular Function Term**	***p*-Value**	***q*-Value**
RNA binding (GO:0003723)	1.00 × 10^−6^	1.87 × 10^−4^
aminoacyl-tRNA ligase activity (GO:0004812)	6.00 × 10^−6^	5.50 × 10^−4^
cholesterol binding (GO:0015485)	3.91 × 10^−4^	2.36 × 10^−2^
sterol binding (GO:0032934)	7.84 × 10^−4^	3.55 × 10^−2^
**GO Cellular Component Term**	***p*-Value**	***q*-Value**
small-subunit processome (GO:0032040)	5.00 × 10^−6^	6.48 × 10^−4^
cytolytic granule (GO:0044194)	1.10 × 10^−5^	6.72 × 10^−4^
intracellular non-membrane-bounded organelle (GO:0043232)	1.03 × 10^−3^	4.20 × 10^−2^
nucleolus (GO:0005730)	1.70 × 10^−3^	4.80 × 10^−2^
nuclear lumen (GO:0031981)	1.97 × 10^−3^	4.80 × 10^−2^

## Data Availability

All source sequencing data for this study is openly available at time of publication at NCBI Sequence read archive (SRA) accession: PRJNA976295.

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
