# Peer review of "Comparison of Transcriptomic Changes in Survivors of Exertional Heat Illness with Malignant Hyperthermia Susceptible Patients"

_ijms, 2023, doi:10.3390/ijms242216124_

Round 1

Reviewer 1 Report

Comments and Suggestions for Authors

The authors present a gene expression study based on blood samples from people suffering from exercise induced hyperthermia. The study is sound and of interest, both for athletics and for medical sciences.

My main "complaint" is that the study is in blood and does not use muscle, which would be the source of the heat generated by the exercise. I do realize the challenges involved with repeated muscle biopsies and do understand that the study is based on blood samples, but in a perfect world it would have been better with muscle samples.

Major comment:

I find the discussion of what all these detected changes really mean to be quite basic and would like some more context. An example is lines 222-4 where references are given to related studies, but not how they fit together and what the reader should learn. Another example is the results in Table 1 and the mention on line 97-8 that there are only downregulated pathways - mostly related to RNA etc. This is not really discussed. Why is it like this?  Is activity supposed to cause RNA to Cajal bodies and because this is down it does not work? Do the authors have anything to say about this result?

Minor comments:

line 359: "If" should be "if"

line 301: I'm not sure what the "it" in "evidence that it lacks" refers to. The sentence is hard for me to understand and I do not get the part about complete sensitivity in this context. Can this be rephrased to be easier to understand?

In conclusion I like this manuscript. It is an interesting topic and of value for the community, but as it is now it reads a bit like a long list of "this is up, this is down". I miss a bit more on what these changes actually mean for our understanding - what is the take-home message? Is the intolerance a result of over-heating or under-cooling? And what should we look for in patients to predict if they are susceptible or not?

Author Response

Reviewer 1

The authors present a gene expression study based on blood samples from people suffering from exercise induced hyperthermia. The study is sound and of interest, both for athletics and for medical sciences.

My main "complaint" is that the study is in blood and does not use muscle, which would be the source of the heat generated by the exercise. I do realize the challenges involved with repeated muscle biopsies and do understand that the study is based on blood samples, but in a perfect world it would have been better with muscle samples.

We wish to thank the reviewer for their assessment of our study. We agree that using muscle biopsies would be ideal for the exploration of muscle related changes. However, due to the challenges of muscle sample acquisition, we wanted to investigate whether it is possible to detect discernible changes in gene expression between phenotypes using a less invasive sample type such as blood. An example of this concept applied to muscle-related disorders is the recent assessment of peripheral blood gene expression in the study of Duchenne’s Muscular dystrophy, which was shown to be useful in the monitoring of disease progression and treatment responses (Signorelli et al. 2021).

Major comment:

I find the discussion of what all these detected changes really mean to be quite basic and would like some more context. An example is lines 222-4 where references are given to related studies, but not how they fit together and what the reader should learn. Another example is the results in Table 1 and the mention on line 97-8 that there are only downregulated pathways - mostly related to RNA etc. This is not really discussed. Why is it like this?  Is activity supposed to cause RNA to Cajal bodies and because this is down it does not work? Do the authors have anything to say about this result?

We thank the reviewer for this constructive feedback, and we have expanded parts of the discussion, providing more context.

Line 210-218, outlines how we believe the elevated gene expression affected EHI cases physiologically.

Line 219-227, links the downregulated telomerase RNA ontology terms to immune response pathways, and added two references; number 15 and 25.

Line 297-307 has been moved to a more appropriate location in the test with more context linking studies together.

Minor comments:

line 359: "If" should be "if"

Changed, now on line 377

line 301: I'm not sure what the "it" in "evidence that it lacks" refers to. The sentence is hard for me to understand and I do not get the part about complete sensitivity in this context. Can this be rephrased to be easier to understand?

Now on line 312,“The sensitivity and specificity of the HTT have not been established due to the absence of a validated diagnostic test for EHI susceptibility, although there is evidence that it lacks complete sensitivity. There have been examples of EHI military personnel who effectively thermoregulate during assessment but subsequently experience a further episode of EHI and are later shown to have an abnormal IVCT response (Gardner et al. 2020).”

Sensitivity in this context refers to the ability of the HTT to detect true positive cases of EHI. An example of incomplete sensitivity is provided in the Gardner et al 2020 study referenced, which reports cases of military personnel passing the HTT but later shown to have episodes of EHI.

In conclusion I like this manuscript. It is an interesting topic and of value for the community, but as it is now it reads a bit like a long list of "this is up, this is down". I miss a bit more on what these changes actually mean for our understanding - what is the take-home message? Is the intolerance a result of over-heating or under-cooling? And what should we look for in patients to predict if they are susceptible or not?

We have modified the concluding paragraph, Lines 338-346, which hopefully better summarises our main findings and provides a more concise take home message.

References:

Signorelli M, Ebrahimpoor M, Veth O, Hettne K, Verwey N, García-Rodríguez R, Tanganyika-deWinter CL, Lopez Hernandez LB, Escobar Cedillo R, Gómez Díaz B, Magnusson OT, Mei H, Tsonaka R, Aartsma-Rus A, Spitali P. Peripheral blood transcriptome profiling enables monitoring disease progression in dystrophic mice and patients. EMBO Mol Med. 2021 Apr 9;13(4):e13328. doi: 10.15252/emmm.202013328. Epub 2021 Mar 10. PMID: 33751844; PMCID: PMC8033515.

Reviewer 2 Report

Comments and Suggestions for Authors

Dear Editor,

Chang et al compared the transcriptome profile of subjects with exertional heat illness, malignant hyperthermia susceptibility by determining whole blood gene expression.

While the manuscript contains quite a lot of information, the significance of these was not made clear for the reader as they are not discussed in a wider pathophysiological context. One important question that should be answered is that What consequences are expected from the altered gene expression?

Another concern about the study design is raised by the fact that intense exercise causes muscle injury, releasing muscle mRNA in the blood. This makes the reader wondering whether the altered expressions correlated with the creatine kinase levels of the plasma?

Another question that comes up in the mind of an average reader is that what functional-biochemical consequences can be expected due to the detected changes in expression? How will these change the outcome of the disease? Will the symptoms of a patient who undergoes multiple heat stroke episodes be different after the events compared to others, who never experienced the severe manifestation of the disease? In other words, how these expressional alterations are expected to change the future condition of the patient? Are there data in the literature which can be explained by the present results? Without answering these "what if" questions, the presented information remains bare data, which may be useful in the future, but standing alone, fail to help our understanding the pathology of these muscle diseases. All in all, I believe that the scientific value of the manuscript could be improved by doing a quasi- meta analysis, that combines the results of multiple previous results (serology, blood cell counts, patients follow-up studies), and correlate with the present results.

Author Response

Reviewer 2

Dear Editor,

Chang et al compared the transcriptome profile of subjects with exertional heat illness, malignant hyperthermia susceptibility by determining whole blood gene expression.

While the manuscript contains quite a lot of information, the significance of these was not made clear for the reader as they are not discussed in a wider pathophysiological context. One important question that should be answered is that What consequences are expected from the altered gene expression?

We suspect that the elevated gene expression in pro-inflammatory pathways at baseline pre-disposes EHI patients to excessive inflammatory responses following intense exercise and heat. I have included a sentence in the discussion highlighting this. Line 215-218

Another concern about the study design is raised by the fact that intense exercise causes muscle injury, releasing muscle mRNA in the blood. This makes the reader wondering whether the altered expressions correlated with the creatine kinase levels of the plasma?

This is an interesting suggestion but we are confident that muscle mRNA released during exercise does not explain our findings. We found the most interesting differences in expression between groups at baseline, i.e., before exercise. There was no significant difference in baseline CK between groups (although individuals in the EHI and MHS groups had high values) or in serum myoglobin (a more sensitive marker of acute muscle injury than CK). There was a greater increase in serum myoglobin (but not CK) with exercise in the MHS group than the other two groups but this was not associated with any phenotype-specific differential effect of exercise on expression after the HTT.

Another question that comes up in the mind of an average reader is that what functional-biochemical consequences can be expected due to the detected changes in expression? How will these change the outcome of the disease? Will the symptoms of a patient who undergoes multiple heat stroke episodes be different after the events compared to others, who never experienced the severe manifestation of the disease? In other words, how these expressional alterations are expected to change the future condition of the patient? Are there data in the literature which can be explained by the present results? Without answering these "what if" questions, the presented information remains bare data, which may be useful in the future, but standing alone, fail to help our understanding the pathology of these muscle diseases. All in all, I believe that the scientific value of the manuscript could be improved by doing a quasi- meta analysis, that combines the results of multiple previous results (serology, blood cell counts, patients follow-up studies), and correlate with the present results

We thank the reviewer for these comments that are similar to the major comment of Reviewer 1 and we now address the translational relevance of our data in the new material in lines 210-218 and lines 297-307. Our discussion includes interpretation of our results in light of previous studies but heterogeneity between patient cohorts and study methodologies precludes anything other than a narrative synthesis.